# Ectopic Expression of *Plasmodium vivax vir* Genes in *P. falciparum* Affects Cytoadhesion via Increased Expression of Specific *var* Genes

**DOI:** 10.3390/microorganisms10061183

**Published:** 2022-06-09

**Authors:** Torben Rehn, Pedro Lubiana, Thi Huyen Trang Nguyen, Eva Pansegrau, Marius Schmitt, Lisa Katharina Roth, Jana Brehmer, Thomas Roeder, Dániel Cadar, Nahla Galal Metwally, Iris Bruchhaus

**Affiliations:** 1Bernhard Nocht Institute for Tropical Medicine, 20359 Hamburg, Germany; torben-rehn@gmx.de (T.R.); pedrolubiana@hotmail.com (P.L.); trangnguyen220399@gmail.com (T.H.T.N.); epansegrau@gmail.com (E.P.); marius.schmitt@t-online.de (M.S.); lisakatharinaroth@gmail.com (L.K.R.); jana.brehmer@web.de (J.B.); cadar@bnitm.de (D.C.); metwally@bnitm.de (N.G.M.); 2Molecular Physiology Department, Zoological Institute, Christian-Albrechts University Kiel, 24118 Kiel, Germany; troeder@zoologie.uni-kiel.de; 3Airway Research Center North (ARCN), German Center for Lung Research (DZL), 24118 Kiel, Germany; 4Department of Biology, University of Hamburg, 22601 Hamburg, Germany

**Keywords:** *Plasmodium falciparum*, *Plasmodium vivax*, VIR, *Pf*EMP1, cytoadhesion, endothelial cell receptor

## Abstract

*Plasmodium falciparum*-infected erythrocytes (*Pf*IEs) adhere to endothelial cell receptors (ECRs) of blood vessels mainly via *Pf*EMP1 proteins to escape elimination via the spleen. Evidence suggests that *P. vivax*-infected reticulocytes (*Pv*IRs) also bind to ECRs, presumably enabled by VIR proteins, as shown by inhibition experiments and studies with transgenic *P. falciparum* expressing *vir* genes. To test this hypothesis, our study investigated the involvement of VIR proteins in cytoadhesion using *vir* gene-expressing *P. falciparum* transfectants. Those VIR proteins with a putative transmembrane domain were present in Maurer’s clefts, and some were also present in the erythrocyte membrane. The VIR protein without a transmembrane domain (PVX_050690) was not exported. Five of the transgenic *P. falciparum* cell lines, including the one expressing PVX_050690, showed binding to CD36. We observed highly increased expression of specific *var* genes encoding *Pf*EMP1s in all CD36-binding transfectants. These results suggest that ectopic *vir* expression regulates *var* expression through a yet unknown mechanism. In conclusion, the observed cytoadhesion of *P. falciparum* expressing *vir* genes depended on *Pf*EMP1s, making this experimental unsuitable for characterizing VIR proteins.

## 1. Introduction

*Plasmodium vivax* is the second most common malaria pathogen in the world with about 4.5 million malaria cases annually. This parasite is particularly widespread in Central and South America and in Asia. However, while 627,000 people die from infection with *P. falciparum*, only about 30,000 people per year die from infection with *P. vivax* [1]. The reasons for the large differences in mortality depend on the difference in infection modes. *P. vivax* infects only reticulocytes, which constitute 0.5–2.0 percent of peripheral blood cells, while *P. falciparum* infects erythrocytes, allowing for much higher parasitemia [2,3]. Furthermore, erythrocytes infected with trophozoite stage *P. falciparum* cytoadhere to endothelial cell receptors (ECRs) of blood vessels to escape elimination via the spleen [4,5]. However, this leads to a number of pathological consequences for the human host, which include blockage of capillaries, interruption of blood supply, hypoxia, edema, endothelial dysfunction, inflammation, organ failure, cerebral malaria, coma, and ultimately death [6,7,8,9,10,11].

Since *Pf*IEs cytoadhere once the parasites are in the trophozoite stage, only the ring stages are detectable in the blood of patients infected with *P. falciparum*. In contrast, all erythrocytic developmental stages (ring, trophozoite, and schizont) can be detected in the blood of patients with a *P. vivax* infection [12,13,14]. Thus, *P. vivax*-infected reticulocytes (*Pv*IRs) are also able to pass through the spleen, as *P. vivax* alters *Pv*IRs to make them more deformable. *P. falciparum* infection, on the other hand, leads to the stiffening of the erythrocyte membrane [15,16].

The cytoadhesion of *P. falciparum*-infected erythrocytes (*Pf*IEs) is mediated by proteins of the *P. falciparum* erythrocyte membrane protein 1 (*Pf*EMP1) family. *Pf*EMP1s are encoded by approximately 60 *var* genes per parasite genome, but in the ring stage parasite, only one *var* is expressed at a time, resulting in only a specific *Pf*EMP1 population being presented on the surface of the trophozoite stage *Pf*IEs [17,18,19]. In total, more than 20 ECRs, such as the endothelial protein C receptor (EPCR), CD36 and intercellular adhesion molecule 1 (ICAM-1), and non-ECRs such as chondroitin sulfate A (CSA), are described to bind *Pf*IEs [20,21,22,23,24,25].

Since all all developmental stages can be detected in the blood of individuals infected with P. vivax, the infection results in only mild symptoms, and *P. vivax* lacks PfEMP1 proteins, it has long been assumed that PvIRs do not cytoadhere to ECRs. However, Carvalho and colleagues demonstrated for the first time that *Pv*IRs are able to adhere to human lung endothelial cells (HLECs), Saimiri brain endothelial cells (SBECs), placental cryosections, CSA, and ICAM-1. Binding to HLECs and SBECs was significantly inhibited by the addition of both trypsin and soluble CSA [26]. Similar results were obtained in a study with 21 Colombian isolates, of which cytoadhesion to lung microvascular endothelial cells (HMVEC-L) was demonstrated for 13 isolates, which could be inhibited by about 50% by an α-ICAM-1 antibody [27]. In another study in which 59 isolates were examined, rosetting was detected in 64%. In addition, 15% of the isolates adhered to CSA, 12% to ICAM-1, and 9% to placental cryosections [28]. However, there were also contradictory results. Thus, in one study, cytoadhesion of *Pv*IRs to CSA and additionally to hyaloronic acid (HA) could be demonstrated in 33 *P. vivax* isolates, but binding to ICAM-1 could not be confirmed [29]. Interestingly, none of the studies showed specific binding to CD36, the main receptor of *Pf*IEs [18,30,31].

Moreover, it has been demonstrated that the amount of circulating schizont stage *Pv*IRs does not correlate with the amount of circulating ring or trophozoite stage *Pv*IRs [12,13]. To this end, the activity of two lactate dehydrogenases was measured and compared with the previously determined parasitemia. This showed the activity of the enzymes was higher than the parasitemia allowed, indicating the presence of non-circulating *Pv*IRs [13]. The hypothesis of a tissue reservoir of *Pv*IRs most likely located in the haemotopoietic niche of the bone marrow and spleen is also supported by the fact that total parasite biomass is a better indicator of disturbances in host homeostasis than peripheral parasitemia [32,33,34].

However, the question arose of which ligands of the parasite mediate binding to the ECRs, since as already mentioned, *P. vivax* does not have *Pf*EMP1-encoding *var* genes. Carvalho and colleagues provided the first evidence that the ligands mediating cytoadhesion of *Pv*IRs to ECRs could be variant interspersed repeat (VIR) proteins. Among others, they demonstrated the binding to HLECs could be inhibited by VIR-specific antibodies [26]. With the *vir* multigene family, *P. vivax* has a gene family belonging to the *pir* (*Plasmodium* interspersed repeats) gene family. Several *Plasmodium* species possess *pir* genes, such as the *rif* of *P. falciparum* or *kir* of *P. knowlesi*. The roles of *pir* genes are not fully understood, but it has been postulated that they are involved in signaling as well as transport and adhesion [35]. In the most recent in silico analysis of the *P. vivax* SaII genome, 295 *vir* genes were annotated, which were grouped into ten subclasses [36]. The length of the *vir* genes varied between 156–2316 bp, and they had between 1–5 exons. Furthermore, in silico analysis found a region encoding for a transmembrane domain in 171 *vir* genes and a motif in 160 *vir* genes that had similarity to the export signal from *P. falciparum* [37,38]. This suggested that the VIR proteins adopt different localizations and thus different functions in the *Pv*IR [38,39]. This was also indicated by the simultaneous expression of several *vir* genes, in contrast with the mutually exclusive expression of *var* genes. Therefore, the VIR proteins do not appear to be involved in antigenic variation [19,38,40,41,42].

To analyze the function of VIR proteins in cytoadhesion, transgenic *P. falciparum* cell lines were generated, expressing three different *vir* genes [39]. Export was predicted for all three encoded VIR proteins (PVX_102635 (VIR10), PVX_108770 (VIR14), and PVX_112645 (VIR17-like)). PVX_108770 and PVX_102635 could be localized to the erythrocyte membrane, but for PVX_108770, only cytoadhesion to CD36, ICAM-1, E-selectin, and VCAM-1 could be detected under static conditions and only to ICAM-1 under flow conditions. Furthermore, only binding to ICAM-1 could be specifically inhibited with antibodies against conserved VIR domains [39]. The results were verified by Fernandez-Bercerra et al., who showed that the VIR protein PVX_108770 adhered particularly to fibrocyte cells expressing ICAM-1 [43].

In order to identify further VIR proteins mediating cytoadhesion to ECRs in our study, in addition to the described VIR protein PVX_108770, further transgenic *P. falciparum* cell lines expressing different *vir* genes were generated. For this purpose, 12 *P. falciparum* transfectants were generated, the localization of the VIR proteins encoded by the expressed *vir* genes was determined, and the binding profile was analyzed using transgenic CHO-745 cells presenting CD36 or ICAM-1 on their cell surface. Five transgenic *P. falciparum* lines showed binding to CD36. However, the transcriptome analyses showed binding is most likely not mediated by the VIR proteins. *Vir* expression leads to a strongly increased expression of specific *var* genes, which encode *Pf*EMP1s of the B- and C-family and could mediate the observed binding phenotype.

## 2. Materials and Methods

### 2.1. P. falciparum Cell Culture

*P. falciparum* isolate 3D7 was cultivated in RPMI 1640 medium (AppliChem, Darmstadt, Germany) containing 10% human serum A+ (Interstate Blood Bank Inc., Memphis, TN, USA) and human O+ erythrocytes (5% haematocrit; UKE, Hamburg, Germany) according to standard procedures [44]. Parasites were synchronized every two weeks using 5% sorbitol solution [45].

### 2.2. Generation of Transgenic P. falciparum Expressing Different vir Genes

The synthetic *vir* genes PVX_050690, PVX_060690, PVX_093715, PVX_101560, PVX_113230, and PVX_115475 were prepared by Invitrogen GeneArt Gene Synthesis (Thermo Fisher Scientific, Waltham, MA, USA) (Appendix A). Subsequent cloning into the pARL1 vector (provided by Tobias Spielmann, BNITM, Hamburg, Germany) was performed by InFusion technique. The synthetic *vir* genes PVX_068690, PVX_077695, PVX_081850, PVX_096925, PVX_097525, PVX_107235, and PVX_108770 were synthesized and cloned into the pARL1-3xHA vector by GenScript (GenScript, Piscataway, NJ, USA) (Appendix A). The pARL1-3xHA vector was generated by inserting a genomic sequence (120 bp) encoding a 3xhemagglutinin-(HA) tag derived from influenza virus A hemagglutinin (accession number: D21182.1) using the *Avr*II and *Xho*I restriction sites.

For transfection, Percoll-purified parasites of the *P. falciparum* isolate 3D7 at late schizont stage were used [46]. Fifty μg of plasmid DNA was used to transfect the parasites by electroporation at 310 V and 950 μF using the NucleofectorTM 2b Device (Lonza, Basel, Switzerland) [47]. Respective 3D7 transfectants (PVX_050690^3D7^, PVX_060690^3D7^, PVX_093715^3D7^, PVX_101560^3D7^, PVX_113230^3D7^, PVX_115475^3D7^, PVX_068690^3D7^, PVX_077695^3D7^, PVX_081850^3D7^, PVX_096925^3D7^, PVX_097525^3D7^, PVX_107235^3D7^, and PVX_108770^3D7^) were selected using 4 nM WR99210 (Jacobus Pharmaceuticals, Plainsboro Township, NJ, USA).

### 2.3. CHO-745 Cell Culture

For the binding assays, transgenic CHO-745 cells presenting CD36 or ICAM-1 on the cell surface were used. For this step, the CHO-745 cells were transfected with the vector pAcGFP-N1 containing either the gene encoding CD36 or the gene encoding ICAM-1. As control, mock-transfected cells were used. The transfection of the CHO-745 cells was performed as described previously [48]. G418 (0.7 mg/mL; Geneticin; Thermo Fisher Scientific, Bremen, Germany) was used as a selection marker. Transfected CHO-745 cells were cultivated in Ham’s F12 medium (Capricorn Scientific, Ebsdorfergrund, Germany) supplemented with 10% heat-inactivated fetal calf serum (Capricorn Scientific, Ebsdorfergrund, Germany) and penicillin-streptomycin (0.1 U/mL; Gibco, Thermo Fisher Scientific, Bremen, Germany). The transgenic CHO-745 cells presenting CD36 or ICAM-1 on the cell surface were routinely sorted for surface expression of the ECRs via fluorescence-activated cell sorting (Appendix A).

### 2.4. Immunofluorescence Assays and Antisera

For the immunofluorescence assays (IFA), blood smears of the respective parasite culture were prepared, air-dried, and then fixed in acetone for 30 min at room temperature (RT). Using a silicone pen (Dakocytomation, Carpinteria, CA, USA), the smear was divided into four equal-sized fields. Each individual field was rehydrated with 100 μL phosphate-buffered saline (PBS; 0.14 M NaCl, 0.3 mM KCl, 8 mM NaH_2_PO_4_, 1.5 mM KH_2_PO_4_, pH 7.4) for 10 min. Next, 60 μL of the respective primary antibody solution diluted in PBS/1% BSA was added to each field and incubated for 2 h in a darkened humid chamber, followed by three washing steps in PBS. The secondary antibodies diluted in PBS/1% BSA, containing 1 μg/mL Hoechst-33342 (Invitrogen, Thermo Fisher Scientific, Bremen, Germany), were applied for 1 h to each field in a darkened humid chamber, followed by 3 washes with PBS. A small drop of Mowiol 4-88 (Calbiochem, San Diego, CA, USA) was added to each field and covered with a coverslip. The smears were dried and stored in the refrigerator until evaluation. The following antibodies were used: α-HA-High Affinity, rat, dilution 1:100 (Hoffmann-LA Roche, Basel, Switzerland); α-skeleton binding protein 1 (SBP1), mouse, dilution 1:1000 (gift from Tobias Spielmann, BNITM, Hamburg, Germany); α-spectrin, rabbit, 1:200 (Sigma-Aldrich, Burlington, MA, USA); *a*-ATS(6H1) (acidic terminal segment), mouse, 1:10,000 (The Walter & Eliza Hall Institute, Parkville, VIC, Australia); α-early transcribed membrane protein 5 (ETRAMP5), rabbit, 1:400 (gift from Tobias Spielmann, BNITM, Hamburg Germany).

### 2.5. Static Binding Assay

A static binding assay was performed to verify the binding capacity of the transgenic parasites expressing different *vir* genes. Highly synchronized trophozoite stage or schizont stage parasites culture were used. Forty-eight hours before the binding experiment, transgenic CHO-745 cells (3 × 10^4^) were seeded as a triplicate on a coverslip (coated before with 1% gelatin) in a 24-well plate. On the day of the assay, the parasite culture was set to a parasitemia of 5% and a hematocrit of 1% in the binding medium (RPMI 1640/2% glucose). To reduce non-specific binding, pre-absorption of *Pf*IEs was performed on mock-transfected CHO-745 cells in a T25 flask for 1 h at 37 °C, with the flask swirled every 15 min. Afterward, the pre-absorbed *Pf*IEs were added to the respective wells with the transgenic CHO-745 cells and incubated for 1 h at 37 °C, again carefully swirling the plate every 15 min. After 1 h, the coverslips were removed from the wells and washed in beakers filled with a binding medium. They were then placed in a new plate that was fixed at a 45° angle, containing 600 μL binding medium in each well, and incubated for 45 min at RT with the overgrown side down. The remaining *Pf*IEs were fixed with 1% glutaraldehyde in PBS for 30 min at room temperature (RT). The fixed cells were stained with a filtered Giemsa/Weisser buffer solution (1:10) for another 30 min. The coverslips were then fixed to a slide with a drop of CV Leica Mounting Solution (Leica, Wetzlar, Germany) with the overgrown side facing downwards and dried overnight. The number of adherent *Pf*IEs was determined by counting the number of bound *Pf*IEs on 300 CHO-745 cells using a light microscope.

### 2.6. RNA Purification and mRNAseq

For RNA isolation, parasites were synchronized 48 h prior to harvest. Ring stage *Pf*IEs were rapidly lysed in a medium 10-times the volume of the cells with pre-warmed 37 °C TRIzol (Invitrogen, Thermo Fisher Scientific, Bremen, Germany) and incubated for 5 min at 37 °C. Samples were stored at −80 °C before RNA was isolated using a PureLink RNA Mini Kit (Thermo Fisher Scientific, Bremen, Germany) according to the manufacturer’s instructions. Contaminations with genomic DNA were removed using the TURBO DNA-free Kit (Invitrogen, Thermo Fisher Scientific, Bremen, Germany) and using the Agencourt RNAClean XP (Beckman Coulter, Krefeld, Germany). The RNA concentration and quality was analyzed with an Agilent 2100 Bioanalyser System using the Agilent RNA 6000 Pico Kit (Agilent Technologies, Ratlingen, Germany).

The total RNA from each sample was prepared for sequencing using the QIAseq Stranded mRNA Library Kit (Qiagen, Hilden, Germany) according to the manufacturer’s instructions. The methodology was based on the 3′-end captures of polyadenylated RNA species and included unique dual indexes (UDIs), which allow direct counting of unique RNA molecules in each sample. Normalized libraries were pooled and sequenced using a 150-cycle (2 × 75 bp paired-end) NextSeq 550 reagent kit v2.5 (Illumina, San Diego, CA, USA) on a NextSeq 550 platform with a depth of 8–16 million paired-end reads generated for each sample. The reads were trimmed and filtered using Trimmomatic [49] and aligned to 3D7 genome data available at PlasmoDB, release 54 [50] using RSEM [51] and Bowtie2 [52] software. The differential expression was tested using DEseq2 for normalization of the raw reads [53].

### 2.7. In Silico Analyses

The VIR proteins were analyzed for properties of exported proteins using various prediction algorithms: prediction of transmembrane domains: TMHMM-2.0—https://services.healthtech.dtu.dk/service.php?TMHMM-2.0 (9 March 2020) [54] and InterPro–https://www.ebi.ac.uk/interpro/ (11 March 2020); prediction of signal peptides (SP): SignalP4.1—https://services.healthtech.dtu.dk/service.php?SignalP-4.1 (9 March 2020) [55]; prediction of the *P. falciparum* export signal: PEXEL-Motive—3 of 5—http://www.dkfz.de/mga2/3 of 5 (12 March 2020).

## 3. Results

### 3.1. Expression of vir Genes in P. falciparum and Localization of the Respective Encoded VIR Protein

A total of 12 transfectants of the *P. falciparum* isolate 3D7 expressing different *vir* genes were generated (PVX_050690^3D7^, PVX_060690^3D7^, PVX_093715^3D7^, PVX_101560^3D7^, PVX_113230^3D7^, PVX_115475^3D7^, PVX_068690^3D7^, PVX_077695^3D7^, PVX_081850^3D7^, PVX_096925^3D7^, PVX_097525^3D7^, and PVX_107235^3D7^). The selection was based on the determination of different domains using various in silico analyses. Thus, when selecting the VIR proteins, care was taken to ensure that all but one of them had one transmembrane domain and were of different sizes. The VIR protein PVX_050690 does not have a predicted transmembrane domain and thus acted as a control of a non-exported protein. In addition, the VIR proteins were analyzed using the online program 3 of 5 to determine whether they contained the protein export motif PEXEL (RxLxE/Q/D) or a PEXEL-like motif (RxLxx) within the first 70 amino acids. A PEXEL motif was identified in the VIR protein PVX_096925, in PVX_097525, and in PVX_101560. A mature N-terminus (MAQ/MAA/MEE) was present in PVX_081850, PVX_107235, and PVX_115475 (Figure 1).

Immunofluorescence analyses were performed to localize the VIR proteins in the *Pf*IEs. In order to detect the VIR proteins, a 3xHA-tag was fused to their C-terminus. Therefore, two specific first antibodies were always used, one detecting the 3xHA-tag and the other a protein that can be assigned to a specific compartment of the *Pf*IE (α-spectrin—erythrocyte membrane, α-ETRAMP—parasitophorous vacuolar membrane (PVM), α-SBP-1—Maurer’s clefts, or α-ATS—conserved acidic terminal segment of *Pf*EMP1).

As suspected from the missing transmembrane domain, the VIR protein PVX_050690 was not exported, and localization was thus restricted to the parasite (Figure 2, Figure 3 and Appendix A). In all other *P. falciparum* transfectants, an export of the VIR proteins in the *Pf*IEs could be detected. Here, colocalization of the VIR proteins with Maurer’s clefts was found for all transfectants. At the trophozoite stage, association with the surface of *Pf*IEs was detected only for the VIR protein PVX_068690 in 11.7% of the *Pf*IEs examined. This percentage increased to 33.9% in the schizont stage *Pf*IEs. The VIR protein PVX_113230 was detected at the erythrocyte membrane in 49% of the analyzed schizont stage *Pf*IEs, followed by PVX_068690 with 34%, PVX_101560 with 28.3%, and PVX_081850 with 17.5%. For all other transfectants, membrane association could only be detected in 8–17% of the analysed *Pf*IEs, with PVX_115475 only detected in the parasite and Maurer’s clefts (Figure 2, Figure 3 and Appendix A). In addition, the extent of colocalization of VIR proteins with *Pf*EMP1 should be investigated. To localize *Pf*EMP1s, an α-ATS antibody directed against the conserved cytoplasmic ATS domain was used. Unfortunately, for unknown reasons, this antibody did not recognize *Pf*EMP1 presented on the surface of *Pf*IEs; however, *Pf*EMP1 localized in the Maurer’s clefts can be detected. Interestingly, only partial colocalization (50–80%) of *Pf*EMP1 with SBP-1, the Maurer’s clefts marker, was detectable. The VIR proteins PVX_068690 and PVX_077695 showed a 93–97% colocalization with the ATS domain of *Pf*EMP1s, both in the trophozoite and in the schizont stage. For almost all other VIR proteins, a colocalization between 20–55% (with one exception of 80%, PVX_081850) can be found (Figure 2, Figure 3 and Appendix A).

### 3.2. Binding Phenotype of Erythrocytes Infected with P. falciparum Transfectants Expressing Different vir Genes

The next step was to investigate whether the transfectants were able to cytoadhere to ECRs, in this case CD36 and ICAM-1. For the binding experiments, the 3D7 isolate used here to produce the transfectants was very well-suited as it has largely lost the ability to adhere to the ECRs through long-term cultivation. Thus, only an average of 4 and 0.15 3D7-*Pf*IE bound to 100 CD36- and ICAM-1-presenting CHO-745 cells, respectively (Figure 4A,B). First, the binding experiments were performed with parasites that were in the trophozoite stage. Of the twelve transfectants examined, significant binding to CD36 was detected for five (PVX_050690, PVX_060690, PVX_068690, PVX_093715, and PVX_096925). Interestingly, parasites expressing PVX_050690, which encoded the only non-exported VIR protein, also showed binding to CD36. Only parasites expressing PVX_096925 showed significant binding to ICAM-1 (Figure 4A). In the next step, it was investigated whether the observed binding phenotype was also detectable for *Pf*IEs in the schizont stage. Here, too, significant binding to CD36 could be detected for all five transfectants. However, none of the transfectants showed any more binding to ICAM-1 (Figure 4B).

### 3.3. Transcriptome Analysis of Transfectants Showing Binding to CD36

Transcriptome analyses were performed to ensure the expression of the *vir* genes did not affect the gene expression of *P. falciparum*, especially the *var* genes. For this purpose, the expression profile of parasites in the ring stage was analyzed, as the *var* genes were expressed in this stage. It was striking that in all transfectants showing binding to CD36, the expression of at least one *var* gene was strongly increased. For the PVX_050690^3D7^ transfectants, which synthesized the only non-exported VIR protein, an increase from 15 to 5750 normalized reads for the *var* gene PF3D7_0712600 was observed, followed by PF3D7_1255200 with an increase from 1.8 to 640 normalized reads. For PVX060690^3D7^, PF3D7_0712600 showed the largest increase, from 18 to 3340 normalized reads. The second highest increase was observed for *var* PF3D7_0900100, from 1 to 680 normalized reads. For PVX068690^3D7^, the strongest increases in expression were measured for two *var* C family genes (PF3D7_0712600 and PF3D7_0712000, from 16 to 4050 and from 100 to 9200 normalized reads, respectively). In the PVX_093715^3D7^ and PVX_096925^3D7^ transfectants, only one *var* gene (PF3D7_0712600 (from 16 to 1440 normalized reads) and PF3D7_0632800 (from 8 to 3060 normalized reads), respectively) was expressed most strongly (Figure 5, Appendix A).

### 3.4. Expression of PVX_108770 in P. falciparum and Characterization of Respective PVX_108770^3D7^ Transfectants

To test whether the expression of the *vir* genes actually has an influence on the expression of the *var* genes and whether the observed cytoadhesion could thus be due to the correspondingly encoded *Pf*EMP1s, PVX_108770 should be expressed, for which binding to ICAM-1 has been demonstrated [39]. PVX_108770 has a transmembrane domain, and like all other VIR proteins that have a putative transmembrane domain, this protein was detected mainly in Maurer’s clefts (Figure 6A,B). However, in about 30% of *Pf*IEs, an association with the erythrocyte membrane was also found. It was striking that the export occurred relatively late, such that the PVX_108770 protein was mainly detectable in the parasite at the trophozoite stage and only to a small extent in Maurer’s clefts (Figure 6B). In contrast with the study mentioned above, *Pf*IE of the transfectant PVX_108770^3D7^ did not bind to CD36 or ICAM-1 (Figure 6C) [39]. However, the loss of cytoadhesion correlated with the results of the transcriptome analysis. Unlike the transfectants, for which binding mainly to CD36 was detected, no increase in expression of *var* genes could be detected (Figure 6D, Appendix A).

## 4. Discussion

Various studies have shown that *Pv*IRs, just like *Pf*IEs, also have the capacity for cytoadhesion, albeit to a lesser extent. Here, the ECR ICAM-1 appeared to be the most important binding partner. However, binding to CSA and hyaluronic acid (HA) has also been described [26,27,28,29,39]. However, it is not yet fully understood which proteins of *P. vivax* mediate this cytoadhesion. Among others, the members of the VIR protein family come into question. This family consists of about 300 proteins of different sizes and structures. There are members with and without transmembrane domains and with and without a signaling and/or a PEXEL motif. It can therefore be assumed the VIR proteins fulfil different functions [36,37,38,39]. In our study, the importance of the VIR proteins for the cytoadhesion of *Pv*IR should be analyzed in detail.

Although progress continues to be made in developing an in vitro culture for *P. vivax*, only cultures with few replication cycles and very low parasitemia have been obtained, making continuous in vitro culture of *P. vivax* impossible to date [56]. An alternative for analyzing the function of *P. vivax* proteins is the heterologous expression of *P. vivax* genes in *P. falciparum*. For several *P. vivax* proteins, the subcellular localization and function could be investigated in this way. For example, overexpression of the *pvcrt-o* gene, which encodes the *P. vivax* vacuolar membrane transporter protein *Pv*CRT-o, in the *P. falciparum* isolate 3D7 resulted in an approximately twofold increase in chloroquine resistance. Furthermore, *Pv*CRT-o and *Pf*CRT were colocalized [57]. The episomal *P. falciparum* transfection system was also used to study the resistance of *P. vivax* to antifolates. For this purpose, the gene *pvdhfr* and various *pvdhfr*-derived mutants encoding the *P. vivax* dihydrofolate reductase (DHFR) were episomally expressed in an antifolate-sensitive *P. falciparum* line. In this way, the influence of sensitivity to different antifolate drugs could be investigated [58,59].

It was shown that the episomal *P. falciparum* transfection system is also suitable for the expression of *vir* genes and for the subsequent localization and functional analysis of the correspondingly encoded VIR proteins. For these analyses, Bernabeu and colleagues used the classical *P. falciparum* expression vector pARL, as in the study presented here [39]. This vector has been used in a number of studies, particularly for localization studies of *P. falciparum* proteins expressed as GFP- or HA-fusion proteins [60,61,62,63]. Based on the expression of *vir* genes in *P. falciparum*, three VIR proteins could be localized that have either no (PVX_112645), one (PVX_108770), or two (PVX_102635) putative transmembrane domains [39]. As suspected, PVX_112645 was not exported and was localized in the parasite. At the trophozoite stage, PVX_108770 was localized in the parasite near the parasitophorous vacuole membrane, while PVX_102635 was localized in the cytosol of the erythrocyte. At the schizont stage, both proteins could be detected at the erythrocyte membrane [39]. In addition to the three VIR proteins described above, our study aimed to identify additional VIR proteins that can act as ligands in binding to ECRs.

For the VIR protein PVX_050690, which like PVX_112646 has no transmembrane domain, no export of the protein into erythrocytes could be detected. In contrast with the study described above, all VIR proteins investigated in this study containing a transmembrane domain were localized during the trophozoite stage mainly within the parasite and also within Maurer’s clefts. During the schizont stage, the localization of the VIR proteins shifted from the parasite to the Maurer’s clefts. Furthermore, depending on the transfectant, an association with the erythrocyte membrane was observed in 0% to 49% of the *Pf*IEs examined. It is worth noting that the localization of PVX_108770 performed here differs in part from that mentioned above [39]. In both studies, the protein can be detected mainly in the parasite during the trophozoite stage. While Bernabeu et al. detected PVX_108770 at the schizont stage on the erythrocyte membrane, this was only the case in about 30% of the *Pf*EIs examined in this study. However, in almost all erythrocytes infected with PVX_108770^3D7^, the VIR protein was detectable in the parasite and in Maurer’s clefts [39].

In five of the transfectants in this study, binding of *Pf*IEs to CD36 could be detected. However, there was no correlation with the membrane association of the respective VIR proteins. This was particularly striking in this context for PVX_050690. As already mentioned, PVX_050690 had no transmembrane domain and could only be localized in the parasite. Nevertheless, significant binding of the respective *Pf*IEs to CD36 could be detected in both the trophozoite and schizonts stages. Another striking feature was that in all transfectants showing binding to CD36, the expression of at least one *var* gene increased strongly compared with the control, with all correspondingly encoded *Pf*EMP1s belonging to the B or C family. PVX_108770^3D7^ transfectants, on the other hand, showed no significant differences in the *var* gene expression profile compared with the control and, accordingly, no binding to the investigated receptors CD36 and ICAM-1.

In summary, the expression of certain *vir* genes (as shown here for PVX_050690, PVX_060690, PVX068690, PVX 093715, and PVX_096925) appears to influence the expression of some *var* genes by a hitherto unknown mechanism. It is therefore imperative, especially for functional analyses, e.g., with the help of mRNAseq, to exclude an influence of the heterologous expression of *P. vivax* genes in *P. falciparum* on the expression profile and/or metabolism of *P. falciparum*.

## Figures and Tables

**Figure 1 microorganisms-10-01183-f001:**
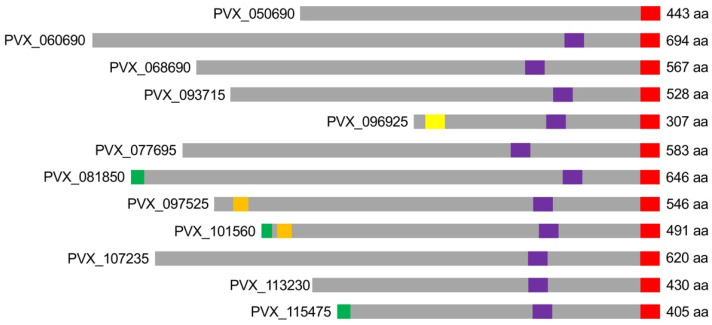
Schematic representation of the VIR proteins whose corresponding encoding *vir* genes were expressed in *P. falciparum* (aa: amino acids; red: HA-tag; purple: putative transmembrane domain; yellow: PEXEL motif; orange: PEXEL-like motif; green: mature N-terminus).

**Figure 2 microorganisms-10-01183-f002:**
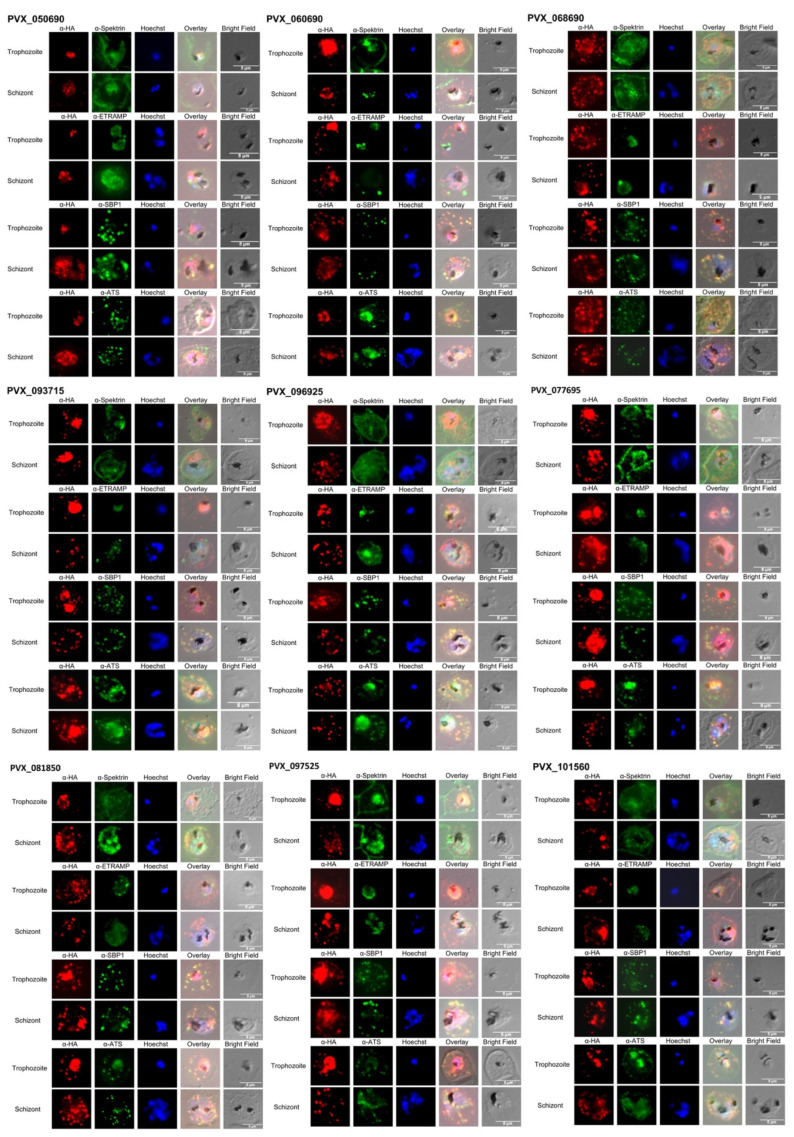
Localization of VIR proteins (PVX_050690, PVX_060690, PVX_068690, PVX_093715, PVX_096925, PVX_077695, PVX_081850, PVX_097525, PVX_101560, PVX_107235, PVX_113230, and PVX_115475) in erythrocytes infected with *P. falciparum* 3D7 transfectants expressing the corresponding *vir* genes. *Pf*IEs were fixed with acetone, and VIR proteins were localized with α-HA (red) in the trophozoite stage and schizont stage *Pf*IEs. Colocalization was performed with α-Spectrin, α-ETRAMP, α-SBP-1, and α-ATS (green). The cell nuclei were stained with Hoechst-33342 (blue). See also Appendix A.

**Figure 3 microorganisms-10-01183-f003:**
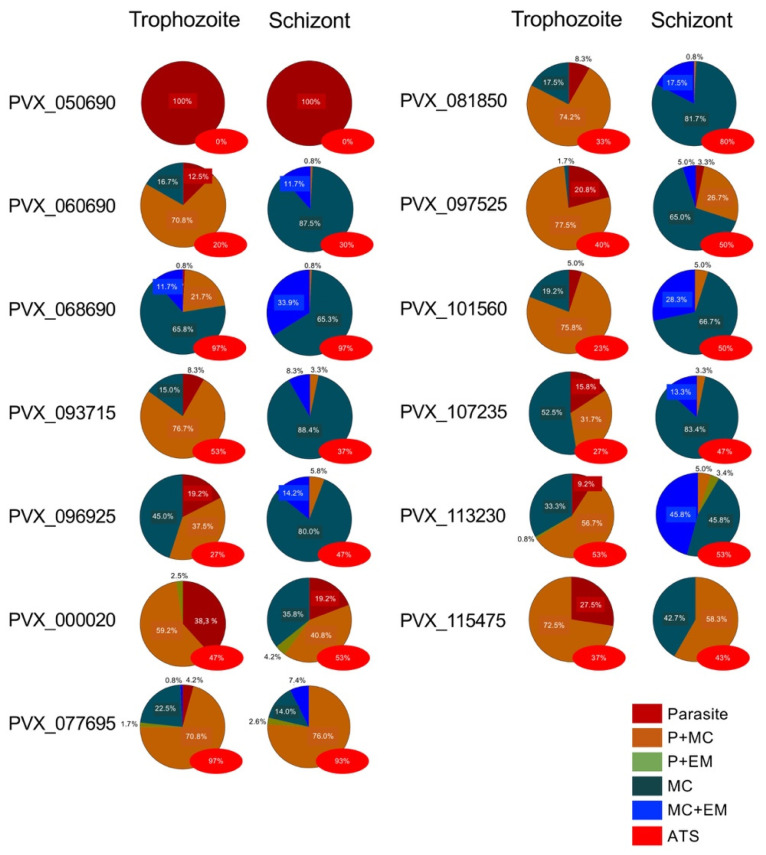
Localization of VIR proteins in erythrocytes infected with *vir*-expressing *P. falciparum* 3D7 transfectants (P: Parasite (dark red), P + MC: Parasite and Maurer’s clefts (orange), P + EM: Parasite and erythrocyte membrane (green), MC: Maurer’s clefts (dark green), MC + EM: Maurer’s clefts and erythrocyte membrane (blue), ATS: Acidic terminal segment of *Pf*EMP1 (red)). For each stage of development (trophozoite and schizont), 120 *Pf*IEs per transfectant were analyzed.

**Figure 4 microorganisms-10-01183-f004:**
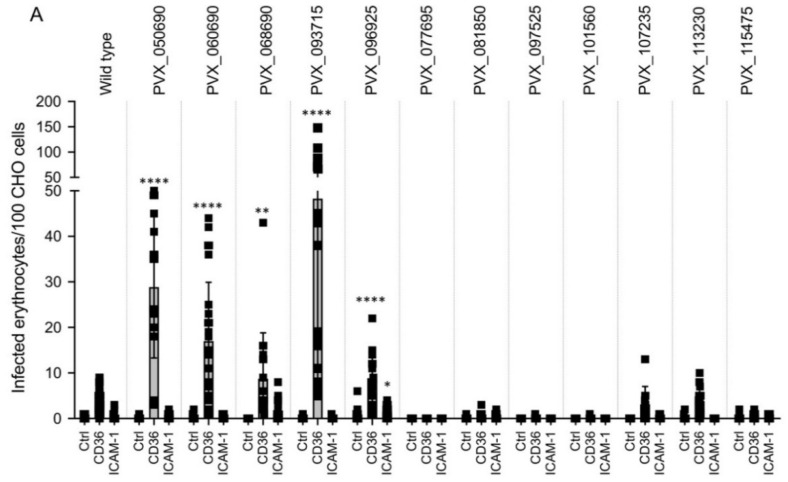
Binding ability of erythrocytes infected with 3D7 (wild type) and 3D7 transfectants (*vir*-expressing) to transgenic CHO-745 cells presenting CD36 and ICAM-1, respectively. (**A**) Binding of trophozoite stage *Pf*IEs to mock (ctrl), CD36-, and ICAM-1-presenting CHO-745 cells. (**B**) Binding of schizont stage *Pf*IEs to mock (ctrl), CD36-, and ICAM-1-presenting CHO-745 cells. The experiments were performed at least three times, and the number of bound *Pf*IEs per 300 CHO-745 cells was determined. Thus, between 9 to 63 coverslips were evaluated per experiment. The unpaired *t* test was used to investigate the statistical significance of the binding of *Pf*IEs to mock-transfected CHO-745 cells (Ctrl) and transgenic CHO-745 cells presenting CD36 or ICAM-1 on their surfaces. (* *p* < 0.05, ** *p* < 0.01, **** *p* < 0.0001).

**Figure 5 microorganisms-10-01183-f005:**
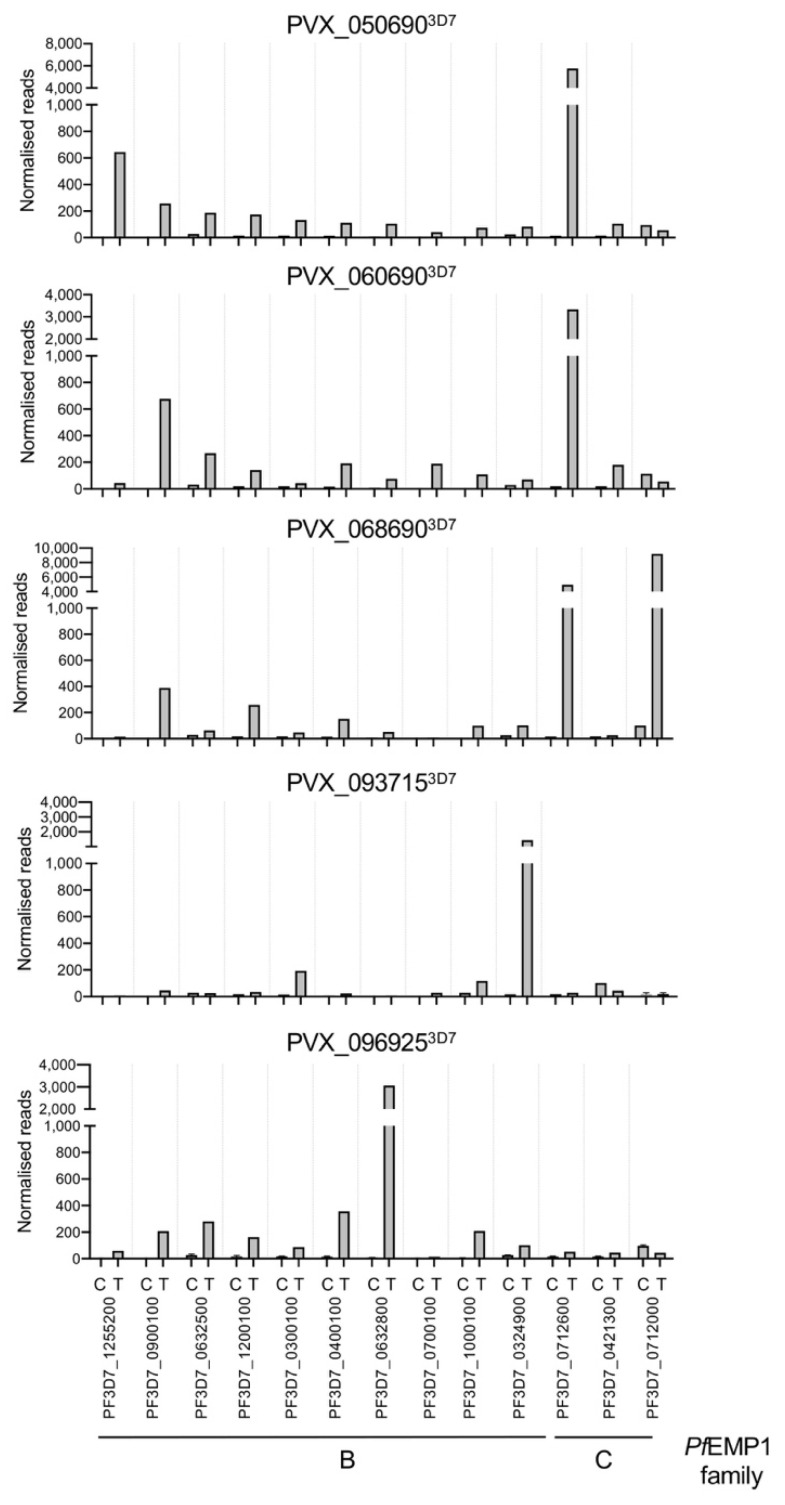
Expression of *var* genes differentially expressed in ring stage *P falciparum* 3D7 transfectants (T) in comparison to non-transfected 3D7 parasites as control (C). The normalized reads are the mean values of two transcriptome analyses derived from two independent biological samples. Exceptions are PVX_093715^3D7^ and PVX_096925^3D7^ with only one biological sample (Appendix A). B: *Pf*EMP1 family B; C: *Pf*EMP1 family C.

**Figure 6 microorganisms-10-01183-f006:**
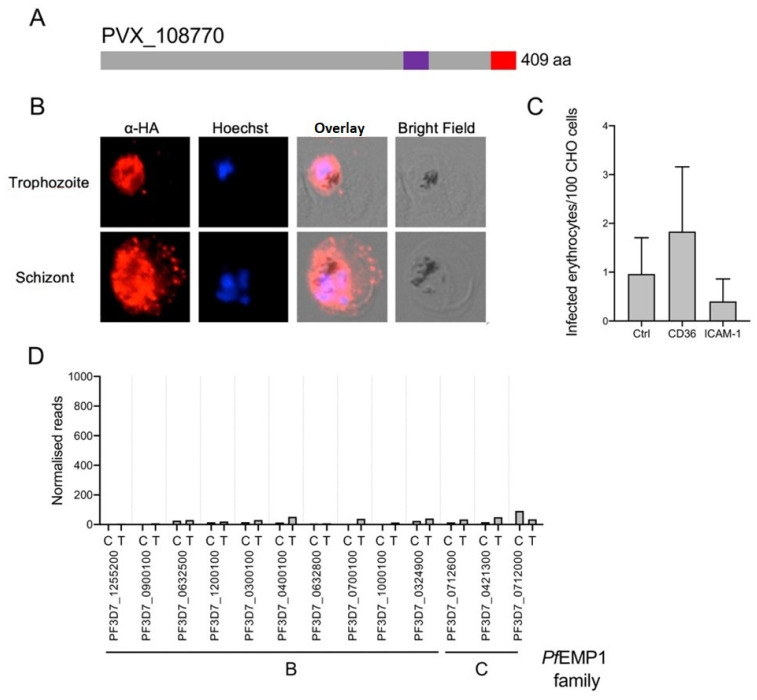
Characterization of PVX_108770^3D7^ transfectants. (**A**) Schematic representation of VIR protein PVX_108770 (aa: amino acids, red: HA-tag, purple: putative transmembrane domain). (**B**) Localization of PVX_108770 in erythrocytes infected with PVX_108770^3D7^ transfectants. *Pf*IEs were fixed with acetone and PVX_108770 localized with α-HA (red) in trophozoite-stage and schizont-stage *Pf*IEs. The cell nuclei were stained with Hoechst-33342 (blue). (**C**) Binding ability of erythrocytes infected with schizont stage PVX_108770^3D7^ transfectants to transgenic CD36- and ICAM-1-presenting CHO-745 cells or mock-transfected CHO-745 cells as control (ctrl). (**D**) Expression of a selected set of *var* genes found to be differentially expressed in various *P. falciparum* 3D7 transfectants in comparison to non-transfected 3D7 parasites as control (T: PVX_108770^3D7^, C: 3D7 wild type) (Appendix A). B:_ *Pf*EMP1 family B; C: *Pf*EMP1 family C.

## Data Availability

Not applicable.

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
