# Peer review of "Ectopic Expression of Plasmodium vivax vir Genes in P. falciparum Affects Cytoadhesion via Increased Expression of Specific var Genes"

_microorganisms, 2022, doi:10.3390/microorganisms10061183_

Round 1

Reviewer 1 Report

Major comments

1. Introduction is too long. Please summarize.

2. Figure 1: The author included 13 P. vivax vir genes in this study. But, PVX_108770 was missed in the Figure 1. This gene also missed in Figures 2, 3, and 4. In stead, the authors separately presented it as different Figure (Figure 6). Did the authors have any reason separated this gene from other genes?

3. Figure 4A: The authors explained that  PVX_096925 showed significant binding to ICAM-1. However, based on the figure, it is not clear whether the value is significant.

4. Figure 4B: Why the authors analysed only 5 transfectants, not all 12 transfectants? If the authors intended to exclude 7 transfectants, please add clear ã…“justifications for the exclusions in the text.

5. Discussion should be rewrite. Please make clear whiat is advances of this study compared to previous studies. What is difference of this manuscript with your or related previously published articles?  And what is the scientific meaning of this study.

6. Many grammar errors and typo-errors were found throughout the manuscript. Please correct all typo-errors. Many pats of this manuscript are not easy to follow. English editing by a native speaker is also appreciated.

Minor comments

1. The authors cited some references as “XXX and colleagues…” in several parts. I am not sure these formats are acceptable in the policy or guideline of Microorganisms, but changing the sentences seem to be more nice for readers.

2. Lines 50-52: Not clear. Please re-write.

3. Line 161: Fifty not 50.

4. Lines 191-197: Please reiwrite to make them clear to understand.

5. Line 203: Please delete “with”

6. Line 325: Please add PVX_ for 068690and 093715.

7. Figure 5. Please change the name for PVX concided with the forms in the text.

8. Line 429: described

9. Line 435: Please add refrence number.

10. Line 451: Please specify the “certain vir genes”.

11. Line 503: Please delete “I”

12. References: Please italicize species name of Plasmodium.

Author Response

Response to Reviewer 1

Major comments

  1. Introduction is too long. Please summarize.

We have shortened the introduction from 2 3/4 to 2 1/4 pages.

  1. Figure 1: The author included 13 P. vivax vir genes in this study. But, PVX_108770 was missed in the Figure 1. This gene also missed in Figures 2, 3, and 4. In stead, the authors separately presented it as different Figure (Figure 6). Did the authors have any reason separated this gene from other genes?

PVX_108770 had already been analysed in detail by Bernabeu et al. (2012), and they identified this protein as a ligand for the endothelial cell receptor ICAM-1. Only after we were able to demonstrate the binding of five of the twelve selected transfectants to CD36 (including a VIR protein localised only in the parasite), we decided to verify our results with PVX_108770. Therefore, we consider it useful to summarize all PVX_108770 results in one figure (Figure 6).

  1. Figure 4A: The authors explained that PVX_096925 showed significant binding to ICAM-1. However, based on the figure, it is not clear whether the value is significant.

Many thanks to the reviewer. We made a mistake in the analysis. The observed binding of PVX_096925 to ICAM-1 is significant, but at p = 0.0134 (*; Mean control: 0.5183; mean ICAM-1: 1.5).

Individual values:

Control: 0, 0, 0, 0, 0, 0, 0, 0, 0, 1, 2, 0, 1, 0, 6, 1, 2, 1);

ICAM-1: 0, 1, 0, 2, 1, 4, 1, 3, 2, 3, 2, 3

We have corrected Figure 4A accordingly.

  1. Figure 4B: Why the authors analysed only 5 transfectants, not all 12 transfectants? If the authors intended to exclude 7 transfectants, please add clear justifications for the exclusions in the text.

It should only be checked whether the binding phenotype is also maintained in P. falciparum infected erythrocytes that are in the schizont stage. A corresponding explanation was added to the result section: “In a next step, it was investigated whether the observed binding phenotype is also detectable for PfIEs in the schizont stage. Here, too, significant binding to CD36 could be detected for all five transfectants.”

  1. Discussion should be rewrite. Please make clear whiat is advances of this study compared to previous studies. What is difference of this manuscript with your or related previously published articles?  And what is the scientific meaning of this study.

The aim of the study was to identify VIR proteins that play a role in cytoadhesion. Since an in vitro cultivation of P. vivax is not yet possible, the present study resorted to the system of heterologous expression in P. falciparum, which is very well established. Bernabeu et al., 2012, also succeeded in identifying interaction partners with the help of this system. Corresponding sentences were added to the discussion section: “In the present study, the importance of the VIR proteins for the cytoadhesion of PvIR should be analysed in more detail”; “In addition to the three VIR proteins described above, the present study aimed to identify further VIR proteins that can act as ligands in binding to ECRs”.

Our results clearly show that ectopic expression regulates the expression of P. falciparum var genes via a yet unknown mechanism. We, therefore, suspect that cytoadhesion is mediated by PfEMP1s, which makes this experiment unsuitable for the characterisation of VIR proteins. Therefore, it was very important for us to report this observation to alert the reader that it is essential, especially for functional analyses, to exclude the influence of heterologous expression of P. vivax genes in P. falciparum on the expression profile and/or metabolism of P. falciparum. The result that leads us to this conclusion is described in particular in the last two paragraphs of the discussion.

  1. Many grammar errors and typo-errors were found throughout the manuscript. Please correct all typo-errors. Many pats of this manuscript are not easy to follow. English editing by a native speaker is also appreciated.

The manuscript has been reviewed by an English-speaking colleague. In addition, a native speaking colleague (Eva Pansegrau) is co-author.

Minor comments

  1. The authors cited some references as “XXX and colleagues…” in several parts. I am not sure these formats are acceptable in the policy or guideline of Microorganisms, but changing the sentences seem to be more nice for readers.

Where possible, however, we have changed the sentence accordingly. However, it is not a question of "xxx and colleagues" being a quotation. The reference is always given in the correct form (number in brackets) at the end of the relevant paragraph/sentence.

  1. Lines 50-52: Not clear. Please re-write.

The sentence was reformulated: “Since PfIEs cytoadhere once the parasites are in the trophozoite stage, only the ring stages are detectable in the blood of patients infected with P. falciparum. In contrast, all erythrocytic developmental stages (ring, trophozoite, schizont) can be detected in the blood of patients with a P. vivax infection”.

  1. Line 161: Fifty not 50.

Corrected.

  1. Lines 191-197: Please reiwrite to make them clear to understand.

Since the paragraph is the listing of antibodies used for immunofluorescence microscopy, it is not clear to us what needs to be rephrased here to improve understanding.

  1. Line 203: Please delete “with”

Corrected.

  1. Line 325: Please add PVX_ for 068690and 093715.

Corrected.

  1. Figure 5. Please change the name for PVX concided with the forms in the text.

Improved Figure 5 was exchanged.

  1. Line 429: described

Corrected.

  1. Line 435: Please add refrence number.

Reference number inserted.

  1. Line 451: Please specify the “certain vir genes”.

Corrected: “In summary, the expression of certain vir genes (as shown here for PVX_050690, PVX_060690, PVX068690, PVX 093715 and PVX_096925) appears to influence the expression of some var genes by a hitherto unknown mechanism.”

  1. Line 503: Please delete “I”

Corrected

  1. References: Please italicize species name of Plasmodium.

Corrected

Reviewer 2 Report

Rehn et al., have studied the effect of P. vivax vir genes in the P. falciparum in terms of cytoadhesion. The work is interesting and well conducted. However, this reviewer has some suggestions to improve the quality of the manuscript. 

Figure 2 should be presented clearly; each protein on a page. 

Figure legend should be elaborated with a clear description.

In each figure legend number of experiments performed should be indicated. 

As CD36 and ICAM-1 look more prominent. Their surface expression should be confirmed with the flow cytometry data. 

Figure 4: "Unpaired t test".... This should be explained clearly... comparison between the groups. 

Author Response

Response to Reviewer 2

Figure 2 should be presented clearly; each protein on a page. 

If we were to present each VIR protein on a single page, we would have 12 pages of figures. Therefore, we have decided to include an appropriate figure to the Supplementary Materials (Figure S3). However, we would like to ask the editors of Microorganisms to let us know what they prefer.

Figure legend should be elaborated with a clear description.

We have also added the name of the VIR proteins examined. But we do not understand what is still missing to understand the Figure.

In each figure legend number of experiments performed should be indicated. 

Corrected:

Figure 3: “For each stage of development (trophozoite and schizont), 120 PfIEs per transfectant were analysed.”

Figure 4: “The experiments were performed at least three times in triplicate and the number of bound PfIEs per 300 CHO-745 cells was determined. Thus, between 9 to 63 coverslips were evaluated per experiment.

As CD36 and ICAM-1 look more prominent. Their surface expression should be confirmed with the flow cytometry data. 

The transgenic CHO-745 cells were regularly sorted for surface expression of endothelial cell receptors via fluorescence-activated cell sorting. This was included in the Materials and Methods section: “The transgenic CHO-745 cells presenting CD36 or ICAM-1 on the cell surface were routinely sorted for surface expression of the ECRs via fluorescence-activated cell sorting.” In addition, a figure (Figure S2) showing the FACS analysis of the transgenic CHO-745 cells was included in the Supplementary Materials.

Figure 4: "Unpaired t test".... This should be explained clearly... comparison between the groups. 

A sentence was included: ”The unpaired t test was used to investigate the statistical significance of the binding of PfIEs to mock-transfected CHO-745 cells (Ctrl) and transgenic CHO-745 cells presenting CD36 or ICAM-1 on their surface.”

Round 2

Reviewer 1 Report

The authors responded properly to my comments and questions. I feel this manuscript can be accepted.

Reviewer 2 Report

The authors have revised the manuscript by considering the reviewer's suggestions. Hereby I endorse the manuscript for publication in its current form.